# Decursinol Angelate Arrest Melanoma Cell Proliferation by Initiating Cell Death and Tumor Shrinkage via Induction of Apoptosis

**DOI:** 10.3390/ijms22084096

**Published:** 2021-04-15

**Authors:** Sukkum Ngullie Chang, Imran Khan, Chang Geon Kim, Seon Min Park, Dong Kyu Choi, Heejin Lee, Buyng Su Hwang, Sun Chul Kang, Jae Gyu Park

**Affiliations:** 1Advanced Bio Convergence Center (ABCC), Pohang Technopark Foundation, Pohang 37668, Korea; sukkumchang@gmail.com (S.N.C.); rjs6538@naver.com (C.G.K.); seonmin@ptp.or.kr (S.M.P.); 2Department of Biotechnology, Daegu University, Gyeongsan 38453, Korea; imranakhan7@gmail.com; 3The Hormel Institute, University of Minnesota, Austin, MN 55912, USA; 4New Drug Development Center, DGMIF, 88 Dongnae-ro, Dong-gu, Daegu 41061, Korea; dongkyu@dgmif.re.kr (D.K.C.); jini150117@gmail.com (H.L.); 5Nakdonggang National Institute of Biological Resources, Sangju 37242, Korea; hwang1531@nnibr.re.kr

**Keywords:** B16F10 murine melanoma cells, decursinol angelate (DA), reactive oxygen species (ROS), apoptosis, cell cycle

## Abstract

Melanoma is known to aggressively metastasize and is one of the prominent causes of skin cancer mortality. This study was designed to assess the molecular mechanism of decursinol angelate (DA) against murine melanoma cell line (B16F10 cells). Treatment of DA resulted in growth inhibition and cell cycle arrest at G0/G1 (*p* < 0.001) phase, evaluated through immunoblotting. Moreover, autophagy-related proteins such as ATG-5 (*p* < 0.0001), ATG-7 (*p* < 0.0001), beclin-1 (*p* < 0.0001) and transition of LC3-I to LC3-II (*p* < 0.0001) were markedly decreased, indicating autophagosome inhibition. Additionally, DA treatment triggered apoptotic events which were corroborated by the occurrence of distorted nuclei, elevated reactive oxygen species (ROS) levels and reduction in the mitochondrial membrane potential. Subsequently, there was an increase in the expression of pro-apoptotic protein Bax in a dose-dependent manner, with the corresponding downregulation of Bcl-2 expression and cytochrome C expression following 24 h DA treatment in A375.SM and B16F10 cells. We substantiated our results for apoptotic occurrence through flow cytometry in B16F10 cells. Furthermore, we treated B16F10 cells with N-acetyl-L-cysteine (NAC). NAC treatment upregulated ATG-5 (*p* < 0.0001), beclin-1 (*p* < 0.0001) and LC3-I to LC3-II (*p* < 0.0001) conversion, which was inhibited in the DA treatment group. We also noticed a systematic upregulation of important markers for progression of G1 cell phase such as CDK-2 (*p* < 0.029), CDK-4 (*p* < 0.036), cyclin D1 (*p* < 0.0003) and cyclin E (*p* < 0.020) upon NAC treatment. In addition, we also observed a significant fold reduction (*p* < 0.05) in ROS fluorescent intensity and the expression of Bax (*p* < 0.0001), cytochrome C (*p* < 0.0001), cleaved caspase-9 (*p* > 0.010) and cleaved caspase-3 (*p* < 0.0001). NAC treatment was able to ameliorate DA-induced apoptosis and cell cycle arrest to support our finding. Our in vivo xenograft model also revealed similar findings, such as downregulation of CDK-2 (*p* < 0.0001) and CDK-4 (*p* < 0.0142) and upregulation of Bax (*p* < 0.0001), cytochrome C (*p* < 0.0001), cleaved caspase 3 (*p* < 0.0001) and cleaved caspase 9 (*p* < 0.0001). In summary, our study revealed that DA is an effective treatment against B16F10 melanoma cells and xenograft mice model.

## 1. Introduction

Melanoma is one of the most devastating types of cancer that occurs frequently, which depends on various environmental factors [1]. Abnormal genetic changes which are caused by environmental factors are responsible for malignant alteration in melanocytes [2]. Skin cancer is differentiated into melanoma and non-melanoma skin cancer occurring worldwide [3]. In the case of skin cancer, melanoma contributes to 75% death, indicating that it is the most aggressive type of skin cancer [4]. Together, genetic and environmental factors are accountable for the progression of melanoma, with additional exposure to UV radiation being the crucial risk factor. Enlightening treatment of melanoma may entail the progress of effective approaches to overcome resistance of different antitumor agents as well as to deactivate related pro-tumoral mechanisms [5,6]. Hence, there is a medical necessity for new nutraceuticals that have the potential for inducing apoptosis in cancer cells with no adverse side effects in normal cells.

Apoptosis, also known as type I programmed cell death (PCD), was first discovered by Kerr [7]. It is specifically characterized by the changes in the biochemical and morphological properties of dying cells, which include cell shrinkage, nuclear condensation and fragmentation, dynamic membrane blebbing and loss of adhesion to the extracellular matrix [8]. The biochemical changes observed during apoptosis are cleavage of chromosomal DNA into internucleosomal fragments, phosphatidylserine externalization and cleavage of several intracellular substrates by specific proteolysis [9,10]. Apoptosis is commonly observed in most cancers and other diseases [11,12]. The two core pathways that function to activate PCD are the intrinsic mitochondrial pathway and the extrinsic death receptor pathway [13]. In the intrinsic pathway, the downregulation of Bcl-2 protein expression is followed by elevated expression of Bax, resulting in increased permeability of the outer mitochondrial membranes, enabling the release of cytochrome C (cyt C) into the cytosol. Cyt C then recruits Apaf-1 and pro-caspase-9 to compose the apoptosome and the downstream trigger of the caspase 9/3 signaling cascade, thereby activating mitochondrial-mediated apoptosis [14]. Chemotherapeutic drugs commonly used for treating cancer cells are responsible for causing apoptosis as well as severe side effects. Moreover, chemotherapeutic drugs do not achieve satisfactory therapeutic results and lead to apoptosis-resistant cancer cells. To overcome this, many phytochemicals are available with a wide range of pharmacological properties.

In a normal cell, proliferation is governed by strict cell cycle phases. The deregulation of these checkpoints may transform a normal cell into a cancer cell. The development of the cell cycle follows four different stages via activation or inactivation of cyclin-dependent kinases, such as serine–threonine kinases upregulated or downregulated depending on the cellular status or the different environmental conditions. Cyclin-dependent kinase-1 (CDK-1), CDK-2, CDK-4 and CDK-6 are required for the initial phase of the G1 to S phase. The complexes like the cyclin D-CDK complexes are vitally important for the progression of the cell cycle. Once the cell has entered the synthesis phase, the cyclin D-CDK complexes are not required and cyclin E and CDK-2 are crucial for the G1 to S phase transition; the cyclin B-CDC2 complex finally initiates the G2 to M phase transition [15]. Various compounds extracted from plants have been reported to target the cyclin-CDK complexes which disrupt the cell cycle.

Autophagy is a highly conserved process crucial for balancing homeostasis in response to nutrient stress. Autophagy is activated in response to different external stimuli and plays an important role in tumor promotion, tumor suppression, removing aggregated misfolded proteins and clearing damaged organelles. Autophagosome formation is dependent on the processing and upregulation of various markers such as ATG5, beclin-1, ATG7 and conversion of LC3-I to LC3-II [16,17]. Autophagy pathway comprises ATG5 and ATG7, which are tangled in the elongation and the termination of the autophagosomal membrane. Moreover, ATG5/ATG7 is also involved in protein light chain 3 (LC3) truncation, lipidation originating from endoplasmic reticulum (ER) membrane or other organelles. Concurrently, beclin-1 is essential for the ATG5/ATG-7 dependent and independent autophagic process. In addition, during autophagy induction, the mammalian target of rapamycin (mTOR) has been found to be inhibited [16]. In our study, we evaluated the concentration of mTOR after drug treatment and also evaluated the conversion of LC3-I to LC3-II.

The Korean Dang-gui or Dong-Quai *Angelica gigas* Nakai (AGN) belongs to the genus *Angelica L.* of the family Umbelliferae [18]. Asian countries such as China, Japan and Korea have been using it as a traditional herbal medicine and food for many centuries. The genus *Angelica L.* is composed of more than 60 species [19,20]. Of these, the coumarin compound decursinol angelate (DA), a derivative of decursin extracted from the roots of *Angelica gigas*, has demonstrated strong anticancer activity [21]. Numerous studies have investigated the role of DA and its isomers’ anticancer activity [22,23,24]. However, more investigations are required to substantiate the previous results and also to explore new molecular targets as DA is a multitargeted therapeutic compound. Our study emphasized the role of DA in inhibiting the growth of murine melanoma B16F10 cells and xenograft mice model.

## 2. Results

### 2.1. Decursinol Angelate Exhibited Anticancer Property on Different Cancer Cell Lines

We evaluated the toxicity level of DA in a dose-dependent (25, 50, 75, 100 μM) manner on various cancer (HepG2, HCT-116, A375.SM and B16F10) cell lines. As shown in Figure 1A, the cell viability at 75 and 100 μM for HepG2 was 58.5%, 44%, HCT-116 (60%, 47.5%), A375.SM (57.4%, 37.29%) and B16F10 (49%, 31%), respectively. The cytotoxicity of DA against B16F10 cells was found to be higher in comparison to other tested cancer lines. The IC_50_ of B16F10 cell line was approximately 75 μM, which was more effective in comparison to other treated cancer cell lines. The percentage of lactate dehydrogenase (LDH) released from the cancer cells was also higher in B16F10 cells in comparison with other cancer cell lines treated with equivalent concentration (Figure 1B). Further, to confirm our observation, we analyzed the morphologic characteristics of all the cancer cell lines (as mentioned above). DA was able to remarkably change the cellular morphology of the studied cancer cells at 75–100 μM, observed through a sharp dose-dependent distortion in the morphology of cells after 24 h treatment (Figure 1C). Through the image analysis and thoroughly assessing the MTT and LDH assay results (Figure 1A,B), we selected 25, 50 and 75 μM for all further experiments against B16F10 cell line.

### 2.2. Decursinol Angelate Induced Cell Cycle Arrest in Murine Melanoma Cells

In order to confirm the antiproliferative effect of DA, we performed the clonogenic assay by exposing B16F10 melanoma cells to varying concentrations of DA (25, 50 and 75 μM). DA was able to inhibit B16F10 cell proliferation in a dose-dependent manner (Figure 2A). As we know, G1/S transitions are regulated by strict restriction points (R point), and when the cell passes by through this point, it will no longer require agents for triggering mitosis (mitogens) to complete DNA replication and the cell cycle. The cyclin-dependent kinases (CDKs) are important kinases that control cell cycle progression by binding to its catalytic partner. Cyclin E is one such that is essentially needed for the progression through the G1 phase and for initiating the replication of DNA through the interaction with the catalytic partner known as cyclin-dependent kinase-2 (CDK2) [15]. Western blot analysis further validated the suppression and downregulation of cyclin D1 (*p* < 0.0001) as well as cyclin E (*p* < 0.0001) dose dependently (Figure 2B) as well as a dose-dependent downregulation in the concentrations of cyclin-dependent kinases such as CDK2, CDK4 and CDK5 and upregulation of p21 CDK inhibitor.

### 2.3. Decursinol Angelate Inhibited Autophagosome Formation in Murine Melanoma Cells

Induction of autophagy could be due to nutrient starvation, reduced growth factor signaling, cellular stress and so forth. One of the most important roles of the autophagy pathway is homeostasis. This complex multistep pathway involves more than 32 autophagy proteins and mTOR plays a crucial role as a mediator of growth factor signaling to autophagy and cellular stresses [25,26,27]. Various studies and research on autophagy have indicated its role in promoting cell survival in numerous tumor cells when treated with chemotherapeutical compounds [28]. In our study, we assessed the effects of different concentrations (25, 50 and 75 μM) of DA on autophagy inhibition in B16F10 cells. Immunoblotting data (Figure 3) revealed DA treatment reduced the expression of autophagy-related genes ATG5 (*p* < 0.0001), ATG7 (*p* < 0.0001), beclin-1(*p* < 0.0001) and suppressed the conversion of the very important microtubule-associated protein light chain LC3-I to LC3-II (*p* < 0.0001) in a dose-dependent manner. We also observed an increase in the concentration of mTOR (*p* < 0.0002) and significantly higher expression of p-mTOR (*p* < 0.0001) levels in a dose-dependent manner. These results indicated that DA inhibited autophagosome formation in B16F10 cells.

### 2.4. Decursinol Angelate Induced Reactive Oxygen Species Production, Mitochondrial Membrane Weakening and Apoptosis in Highly Metastatic Human Melanoma and Murine Melanoma Cells

ROS reacts directly with double bonds of polyunsaturated fatty acids, resulting in the production of lipid hydroxides which contribute to the disruption of necessary biomolecules required for biological mechanisms [29,30]. In this study, we found DA to be a potential oxidative stress inducer in A375.SM and B16F10 melanoma cells by generating reactive oxygen species. DA-treated melanoma cells, A375.SM and B16F10 cells had elevated levels of ROS via the electron transport-mediated system (Figure 4A or Figure 5A). We also observed a significant reduction in mitochondrial membrane potential (ΔΨ_m_). The reduction in ΔΨ_m_ is an indication towards early stage apoptosis and is associated with reactive oxygen species production, DNA damage and increased membrane permeability [31]. To ascertain this claim, we performed mitotracker staining (Figure 5B) to validate the damage dealt by DA on mitochondrial membrane potential calculated through the intensity of reduced red fluorescence in a dose-dependent manner. Next, we assessed the induction of apoptosis. DA-treated groups exhibited a dose-dependent increase in nuclear condensation, nuclear shrinkage and pycno-nuclei formation in both cell lines, whereas apoptotic indicators were observed to a lesser extent in the control groups. Through Hoechst staining, we determined the apoptotic index as depicted in Figure 4B or Figure 5C. In addition, AO/EtBr staining was also performed to investigate apoptotic cell formation. Acridine orange (AO) is taken up by the healthy cells and the nucleus appears green, whereas ethidium bromide (EtBr) is taken up by cells that are either dead or distorted and the nucleus appears as orange-red when observed under fluorescence microscope. DA treatment contributed to a dose-dependent formation of apoptotic bodies as observed by the orange-red fluorescent nucleus in A375.SM as well as B16F10 cells (Figure 4C or Figure 5D). Synchronously, there was activation of caspases (caspase-9, caspase-3) and release of cytochrome C.

From the mitochondria to cytosol is required for triggering apoptosis [32]. The most important regulators of mitochondria-mediated apoptosis are caspase-3/9, B-cell lymphoma (Bcl)-2-associated X protein (Bax) and the B-cell lymphoma (Bcl)-2 family [33,34]. Therefore, we also investigated these markers through immunoblotting. DA treatment caused increased Bax (*p* < 0.0001) levels and downregulated Bcl-2 (*p* < 0.0001) and increased cytosolic cytochrome C (*p* < 0.0001) expression in A375.SM cells (Figure 4D) and a similar dose-dependent increase in early apoptosis markers of Bax (*p* < 0.0001) and cytochrome C (*p* < 0.0010) in B16F10 cells as depicted in Figure 6A. Our result was further substantiated by flow cytometry analysis for apoptosis where DA showed dose-dependent increase in apoptotic percentage in B16F10 cells (Figure 6B).

### 2.5. NAC Reversed DA-Induced Autophagy Inhibition and Rescued G1 Cell Cycle Arrest in Murine Melanoma Cells

In order to assess if NAC treatment would have any significant changes on the autophagy and cell cycle arrest induced by DA, we treated B16F10 cells with NAC (5 mM, 3 h) and DA (75 μM, 24 h). Our findings from this experiment were able to further validate that NAC treatment and NAC + DA treatment group saw an increase in autophagy-related protein markers, which was inhibited on the DA treatment group evaluated through ATG-5 (*p* < 0.0001), beclin-1 (*p* < 0.0001) and LC3-I to LC3-II (*p* < 0.0001) from Western blotting analysis (Figure 7A). NAC treatment also reversed the cell cycle arrest induced by DA treatment. We noticed a systematic upregulation of important markers needed for progression to G1 cell phase such as CDK-2, CDK-4, cyclin D1 and cyclin E upon NAC treatment and NAC + DA treatment groups (Figure 7B), which were further downregulated in DA treatment groups. Our findings amply exhibited that NAC was able to reverse autophagy inhibited by DA treatment and G1 cell cycle arrest in B16F10 murine melanoma cells.

### 2.6. NAC Attenuated DA-Induced Mitochondrial-Mediated Apoptosis in Murine Melanoma Cells

NAC (N-acetyl-L-cysteine) is a synthetic antioxidant and a precursor of intracellular glutathione and cysteine, which is involved in the inhibition of ROS-dependent apoptosis [35,36] by mimicking the role of natural antioxidant. The anti-ROS activity of NAC is the reaction of its free radical scavenging property, either directly through the redox potential of thiols, or secondarily by increasing the level of glutathione in cells [37]. Our results indicated increased cell viability in the NAC and NAC + DA group as compared to the DA treatment group (Figure 8A). In addition, we also observed a significant fold reduction (*p* < 0.05) in ROS fluorescent intensity of NAC and NAC + DA-treated group (Figure 8B). In addition, immunoblot analysis also confirmed that NAC significantly reversed the expression of Bax, cytochrome C, cleaved caspase-9 and cleaved caspase-3 (Figure 8C). Our results elucidated that NAC attenuated DA-induced apoptosis in B16F10 murine melanoma cells.

### 2.7. DA Shrunk Tumor Formation in Nude Mice

In order to investigate the inhibitory effects of DA on tumor growth in vivo, B16F10 cells were implanted in the right flanks of nude mice and tumor growth and body weight were recorded daily after systemic treatment with DA. As observed, treatment with DA reduced the volume and weight of tumor in a dose-dependent manner (Figure 9A–C). Additionally, DA exposure also normalized the body weight by reducing the tumor volume (Figure 9D). Histopathological analysis of the tumor revealed apoptotic features in the DA-treated mice, as evidenced by the distorted and condensed nuclei morphology, and the densely packed cells with slight invasion in the surrounding tissues in the control group (Figure 9E). Moreover, observation from the vital organs such as heart, lungs, liver, kidney and spleen revealed no toxic effects or cellular distortion and were normal (Figure 9E). Additionally, immunohistochemistry analysis from tumor sections also supported our finding. The expression of p21 was densely upregulated in the DA-treated tumors in comparison with the normal control tumors, indicating a cell cycle arrest on DA-treated tumors (Figure 10A). The expression of cleaved caspase 3 (Figure 10B) was also markedly higher in DA-treated tumors dose dependently, whereas control B16F10 tumor had lesser expression of these apoptotic markers. These results were further bolstered by TUNEL assay where DA treatment tumor groups saw a significant rise in the apoptotic cells analyzed by browning of the cells (Figure 10C). Western blot results also showed similar findings for the induction of apoptosis as seen through dose-dependent upregulation of Bax (*p* < 0.0001), cyto C (*p* < 0.0001), Bcl-2 (*p* < 0.0001), caspase-3 (*p* < 0.0001) and caspase-9 (*p* < 0.0001) markers and the downregulation of cell cycle markers CDK-2 (*p* < 0.0001) and CDK-4 (*p* < 0.0001) (Figure 10D).

## 3. Discussion

Most of the available drugs for treatment of melanoma are often accompanied by severe side effects. Therefore, there is always this requirement for phytochemical-mediated remedies and nutraceuticals having anticancer potential. Numerous studies have unraveled the anticancer potential of DA and its multi-targeted therapeutic properties. However, to our knowledge, none of the publications elucidated their molecular mechanism in B16F10 cells. We observed that DA increased the nuclear fragmentation and chromatin condensation in a dose-dependent manner which is one of the hallmarks of apoptosis [11]. Despite extensive efforts, we still have much to learn about the molecular mechanisms and the involvement of ROS in apoptosis [35]. ROS production can happen either from mitochondrial electron transport chain complexes or through other means such as superoxide production from the endoplasmic reticulum, as observed in yeast [38]. The production of ROS by mitochondria was first proven in the 1960s (Jensen, 1966). It has been shown that metazoans produce mitochondrial ROS through the production of superoxides from complex I, III and other complexes [39]. This superoxide production leads to hydrogen peroxide formation and other downstream products that cause many non-specific oxidative impairments and damage thereby, contributing to numerous diseases [40,41]. The changes in the ROS level and the association with the altered mitochondrial activity has been found to be caused by the disrupted mitochondrial metabolism such as accumulated NADPH observed in dysfunctional or mutated mitochondria [39]. It is also well known that ROS is a potent inducer of JNK. Further, it is crucial to explore and elucidate the molecular mechanisms of the ROS-mediated JNK activation [42]. We have demonstrated ROS production through fluorescent staining. ROS plays a crucial role in apoptosis by activating numerous apoptotic markers like Bax; upregulated Bax leads to the dissipation of mitochondrial membrane potential (ΔΨ_m_), and membrane weakening results in the release of cytochrome C to the cytosol. This, in turn, activates the proteases such as caspase-9 and caspase-3, thereby inducing apoptotic cell death through the intrinsic pathway. To prove that DA induced oxidative stress-facilitated apoptosis, we used NAC as a ROS scavenger which blocks the ROS production. We observed a reduction in cell death in the DA + NAC-treated group compared to the DA alone group. The results observed through these experiments were consistent with many published reports of inducing mitochondrial-mediated apoptosis by other natural compounds [43]. The in vivo results of B16F10 xenograft in nude mice also illustrated cell cycle arrest and apoptosis, consistent with the in vitro findings. Based on our results, we conclude that DA predominantly induces oxidative burst inside B16F10 cells, triggering numerous cellular actions leading to mitochondria-mediated intrinsic apoptosis (Figure 11). To the best of our understanding, this is the first report in which we show that DA displays substantial apoptotic potential against the murine melanoma B16F10 cells. Owing to the significant anticancer potential of DA, we propose it as a suitable drug for murine melanoma cells and to carry out further experiments.

## 4. Materials and Methods

### 4.1. Chemicals, Antibodies and Reagents

Decursinol angelate (Item no: 25212), (PubChem CID: 776123) was purchased from Cayman (Cayman Chemicals, Ann Arbor, Michigan, United States). DMEM, 3-(4,5-dimethylthiazol-2-yl)-2,5-diphenyltetrazolium bromide (MTT), Hoechst 33342, dichlorodihydrofluorescein diacetate (H_2_DCFDA), acridine orange (AO), dimethylsulfoxide (DMSO) and mitotracker were purchased from Sigma (Sigma-Aldrich, St. Louis, MI, USA). Fetal bovine serum was purchased from Gibco, (Gibco-Thermo Fisher, Waltham, MA, USA). Details of the primary and secondary antibodies used are mentioned in Appendix A. All the other solvents used for the study were of highest grade and supplied by Sigma (Sigma-Aldrich).

### 4.2. Cell Culture and In Vitro Assays

B16F10 cells (passage number 2, ATCC), HepG2 cells (passage number 7, KCLB) were cultured in DMEM and supplemented with 10% fetal bovine serum (Gibco, USA) and 1% penicillin–streptomycin (Gibco, USA). HCT-116 cells (passage number 2, KCLB) were cultured in RPMI-1640 supplemented with 10% fetal bovine serum (Gibco, USA) and 1% penicillin–streptomycin (Gibco, USA). A375.SM (passage number 2, KCLB) was cultured in MEM media supplemented with 10% heat-inactivated FBS and 1% penicillin–streptomycin (Gibco, USA). The cells were maintained in a CO_2_ incubator (5% CO_2_) at 37 °C. For performing the experiment, cells were cultured in a 12-well plate at a density of 1 × 10^5^ cells/well. After the cells had reached 60–75% confluency, cells were treated with 25, 50 and 75 μM DA for Western blotting and staining experiments.

### 4.3. MTT Assay and Morphological Assessment

MTT assay was performed to determine the percentage of cell viability. We referred to the previously published paper [44,45]. B16F10, HepG2, HCT-116, A375.SM cells were seeded in a 96-well flat-bottom microtiter plate at a density of 1 × 104 cells/well in 200 μL DMEM and allowed to adhere for 24 h at 37 °C in a CO_2_ incubator. Once the cells reached 60–70% confluency, the culture medium was replaced with fresh medium. Cells were exposed to varying concentrations of DA (10, 25, 50, 75, 100 μM) and incubated for 24 h at 37 °C. Subsequently, the culture medium was replaced with 100 μL DMEM along with 10 μL of MTT working solution (5 mg/mL in PBS) onto each well; the plates were allowed to incubate for 4 h at 37 °C. After incubation, the medium was aspirated and the formazan crystals were solubilized by adding 50 μL DMSO/well and allowed to incubate for 30 min. Finally, the dissolved formazan crystals (purple color) in DMSO were quantified at 540 nm using an ELISA plate reader. The percentage of cell viability was calculated based on the following formula: Cell viability (%) = (Treated group/control group) × 100%

For determining the morphological characteristics and changes after 24 h exposure to DA (25, 50, 75 and 100 μM), the cells were observed under a compound microscope.

### 4.4. LDH Assay

DA was treated on B16F10 cells to induce and trigger cellular cytotoxicity, which would thereby lead to release of extracellular LDH cytosolic enzyme. LDH assay kit (Sigma-Aldrich) was used to determine the cellular cytotoxicity and the experiment was carried out according to the manufacturer’s instructions.

### 4.5. Clonogenic Assay

The assay was performed as per the protocol mentioned [46,47]. Shortly, B16F10 cells were seeded in DMEM medium in 12-well tissue culture plates (BD Falcon, CA) at a seeding density of 4 × 10^3^ cells/well. After the cells had adhered onto the plates, the cells were exposed to varying concentrations of DA (25, 50 and 75 μM) for 7 days, and were fixed in 6% (w/v) glutaraldehyde, and later on stained with 0.1% (w/v) crystal violet.

### 4.6. Determination of Morphological Changes and Apoptosis by Fluorescence Staining

B16F10 cells were cultured in DMEM and treated with different concentration of DA for 24 h. Next, the cells were washed twice in PBS, and allowed to fix in cold 4% formaldehyde. The cells were again washed with PBS and treated with Hoechst 33342 (1 mg/mL) and incubated at 37 °C for 10 min in the dark [46,47]. After incubation, cells were washed again with PBS and the intensity of the resulting fluorescence was detected using an Olympus BX50 fluorescence microscope. Assays were performed in triplicates for each independent experiment for quantifying the apoptotic index. After staining, we observed around 100 cells from 10 different random microscopic fields. The apoptotic cells were calculated by dividing the total number of cells with distorted apoptotic morphology by the total number of intact unaffected cells and thus multiplying by 100. For counting the apoptotic cells, we used Image J software. Acridine orange (AO) and ethidium bromide (EtBr) were used for detecting the apoptotic cell formations [47,48]. Shortly, after allowing the cells to adhere and reach confluency, the cells were exposed to varying concentrations of DA and incubated for 24 h. Later, cells were washed with PBS and were fixed with cold 4% formaldehyde. The fixed cells were then washed again with PBS and treated with 1:1 ratio of AO/EtBr and incubated for 30 min at 37 °C. The cells were washed again with PBS, and the intensity of the resulting fluorescence was detected using an Olympus BX50 fluorescence microscope.

### 4.7. H_2_DCFDA and Mitotracker Staining

For the detection of intracellular production of reactive oxygen species (ROS), B16F10 cells were incubated with 2,7-dichlorodihydrofluorescein diacetate acetyl ester (H_2_DCFDA, 10 mM) for about 10 min in the dark [48]. Theoretically, when there is the presence of intracellular H_2_O_2_, the non-fluorescent membrane-permeable H_2_DCFDA is converted into impermeable fluorogenic 2′,7′-dichlorofluorescein. The ROS intensity was calculated by using Image J software. Next, we estimated the mitochondrial health by treating cells with mitotracker (5 mM) after necessary DA treatment. Cells were incubated for 10 min. After two washes with PBS, the fluorescent images were captured using an Olympus BX50 fluorescence microscope. The mean fluorescence intensity was analyzed using Image J software.

### 4.8. Flow Cytometry Analysis

For analysis of cell apoptosis by flow cytometry, B16F10 cells were cultured in DMEM and treated with different concentration of DA. After incubation for 24 h, we determined cellular apoptosis using Annexin V/PI apoptosis detection kit following the manufacturer’s protocol.

### 4.9. Isolation of Proteins

For extraction of the whole cellular protein, the cells were lysed by the addition of RIPA buffer (1:100 dilutions with protease and phosphatase inhibitor) followed by incubation on ice for 30 min [49]. Later, the mixture was centrifuged at 12,000 rpm at 4 °C for 15 min. The supernatant was collected in a different centrifuge tube for molecular analysis, and the protein concentration was quantified through Bradford assay.

### 4.10. Western Blotting

For analysis, we extracted the cell lysates (whole-cell) and loaded them on an SDS–polyacrylamide gel and the proteins were allowed to resolve by electrophoresis and transferred to a polyvinylidene difluoride membrane (PVDF) [50]. The blots were immersed in 5% non-fat milk prepared in PBS or 3% BSA in TBST solution and incubated for 1–2 h for blocking. Next, the PVDF membranes were probed with the primary antibodies against a specific protein of interest overnight at 4 °C. This was followed by probing with a secondary antibody for 1–2 h at room temperature. All washes between and after incubations were done thrice using the wash buffer. After the final wash, the PVDF membrane blots were visualized through an enhanced chemiluminescence Western blotting detection reagent (Amersham Biosciences Inc., Piscataway, NJ, USA).

### 4.11. Xenograft Study in Mice

Male BALB/c nude mice 4–6 weeks old were purchased from Orient Bio, Gyeonggi-do, South Korea. Animal experiments were approved by the animal ethical committee of Pohang Technopark Foundation, Republic of Korea and were performed and carried out according to the national guidelines of rules and regulations. The procedures involving animals and their care were confirmed accordingly with the institutional guidelines that are in compliance with national and international laws and policies and with “ARRIVE” guidelines (Animals in Research Reporting in vivo Experiments). The approval number for carrying out the mice experiment is ABCC2018010.

The antitumor potential of DA was evaluated and assessed by inducing tumors in nude mice by subcutaneously injecting 5 × 10^5^ B16F10 cells [51,52,53,54]. DA was administered orally at 100 mg/kg and 200 mg/kg, three times a week. Once the tumor reached a size of 50 mm^3^, the mice were randomly divided into three different groups (*n* = 5/group): group I—control (no DA treatment), group II—100 mg/kg DA and group III—200 mg/kg DA, as per the body weight (BW). Tumor diameters were measured using a digital Vernier caliper every 3 days, and the tumor volume was calculated in mm^3^ using the following formula:Volume = (width)^2^ × length/2

After 4 weeks, the mice were anesthetized using isoflurane liquid inhalation (Wellona Pharma) and sacrificed, and the tumor was removed, collected and processed for histological studies and Western blotting analysis.

### 4.12. H&E and Immunohistochemical Staining

For performing H&E staining, the previously published protocol was followed [55]. Whereas for immunohistochemistry, we refer to our previously published paper [56]. Firstly, the paraffin-embedded slides were heated for antigen retrieval step and dewaxing step with xylene twice (5 min). Next, the slides were dehydrated in 100%, 95%, 80%, 70%, 50% EtOH (5 min). After, the slides were washed with PBS (3 times) and incubated with blocking buffer solution (SuperBlock, Thermofisher Scientific, Waltham, MA, USA). Later, slides were treated with primary antibodies p21, cleaved caspase 3, cytochrome C (Bioworld Technology, St. Louis Park, MN, USA) and incubated for 3 h at RT. After incubating with primary antibody, the slides were again washed with PBS and incubated for 30 min with secondary antibody at RT. After secondary antibody treatment on the slides, a few drops of DAB substrate solution were added to each slide and incubated for a further 15 min. Afterwards, washing step with distilled water and hematoxylin treatment for 3 min to counterstain was performed. Finally, the dehydration process and dewaxing steps were performed and the slides were fixed with a mounting medium, allowed to air dry and observed under Olympus BX50 fluorescence microscope for data collection.

### 4.13. Statistical Analysis

For determining the quantitative results of all the experiments performed in this study, the values are expressed as a mean ± standard deviation (SD) of the experiments. The statistical significance and the differences in the experimental groups were calculated by using one-way analysis of variance (ANOVA) with Tukey’s comparing all pair of columns and Student’s *t*-test from Prism software; where * represents *p*-values < 0.05, ** represents *p*-values < 0.01 and *** represents *p*-values < 0.001.

## Figures and Tables

**Figure 1 ijms-22-04096-f001:**
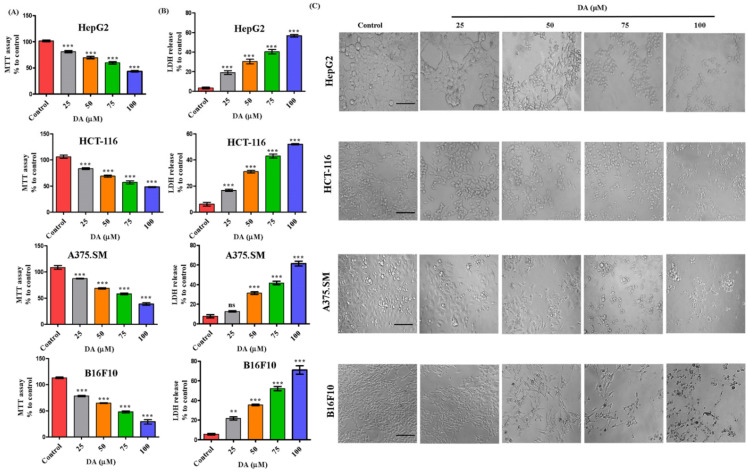
Toxicity study of DA against various cancer cells. (**A**) MTT assay to evaluate the cytotoxicity of DA on various cancer cell lines human liver cancer (HepG2) cell line, human colon cancer (HCT-116) cell line, highly metastatic human melanoma (A375.SM) cell line and murine melanoma (B16F10) cell line. (**B**) LDH assay performed on different cancer cells respectively, after DA (0–100 μM) treatment on cells. (**C**) Morphological changes were captured at various doses with DA (25–100 μM) at 200× magnification (scale bar = 100 µm).

**Figure 2 ijms-22-04096-f002:**
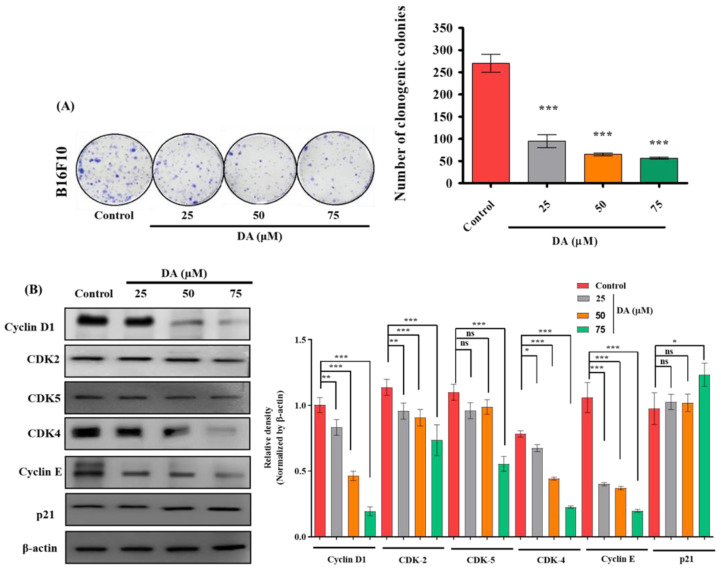
(**A**) Clonogenic assay and quantification of B16F10 cells cultured in the presence and absence of DA over 7 days, followed by crystal violet staining. (**B**) DA was treated to B16F10 cells for 24 h and cell cycle protein levels were detected, such as cyclin D1, CDK2, 5, 4, cyclin E, and p21. Densitometry analysis of the respective proteins was evaluated by Image J software, and results were normalized with β-actin. The data are represented as the means ± standard deviation (SD) of three independent experiments; ns—non-significant; * *p* < 0.05, ** *p* < 0.01, *** *p* < 0.001 vs. control, calculated through ANOVA prism.

**Figure 3 ijms-22-04096-f003:**
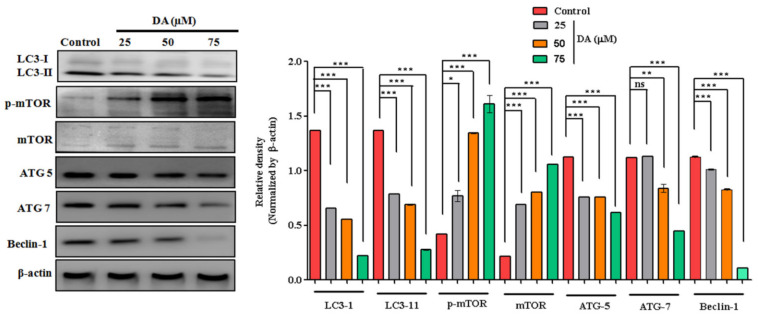
DA inhibited autophagy activation. Twenty-four hours DA treatment on B16F10 cells and evaluation of autophagic protein level such as LC-3I-II, mTOR, p-mTOR (Ser 2448), ATG5, ATG7 and beclin-1. Densitometry analysis of the respective proteins was evaluated by Image J software, and results were normalized with β-actin with respect to controls. The data are represented as the means ± S.D. of three independent experiments * *p* < 0.05, ** *p* < 0.01, *** *p* < 0.001 vs. control, calculated through ANOVA prism.

**Figure 4 ijms-22-04096-f004:**
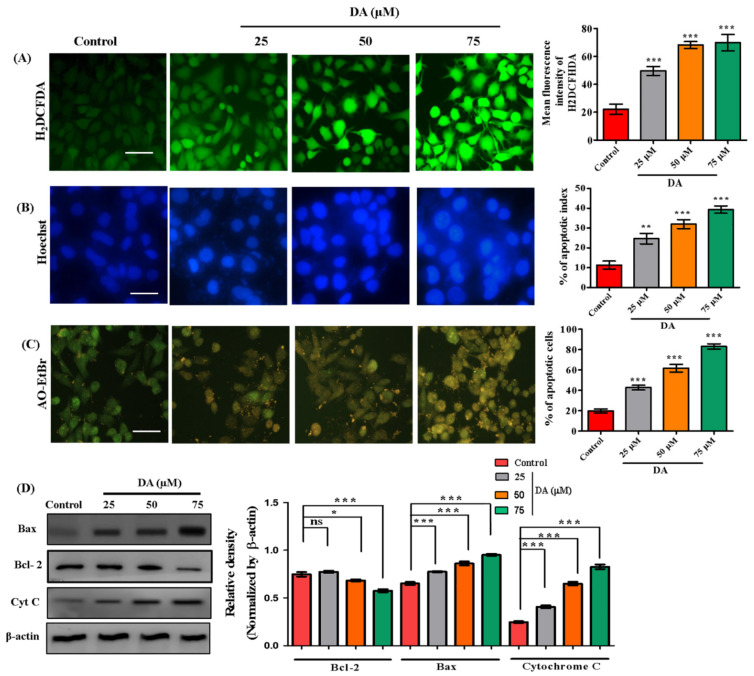
DA induced apoptosis in human melanoma A375.SM cells. (**A**) ROS was evaluated through H_2_DCHFDA (cellular ROS) staining in highly metastatic human melanoma A375.SM cells after 24 h DA treatment. (**B**) Hoechst staining for detection of apoptotic morphology and live-dead cells. (**C**) AO–EtBr staining. (**D**) Evaluation of apoptotic protein levels (Bcl-2, Bax and cytochrome C) by Western blotting. Image quantification was done with the help of Image J software. Apoptotic cells were calculated as percent apoptotic nuclei compared with a total number of cells. Images were taken at 200× magnification using an Olympus BX50 fluorescence microscope (scale bar = 100 µm). The data are represented as the means ± S.D. of three independent experiments * *p* < 0.05, ** *p* < 0.01, *** *p* < 0.001 vs. control, calculated through ANOVA prism.

**Figure 5 ijms-22-04096-f005:**
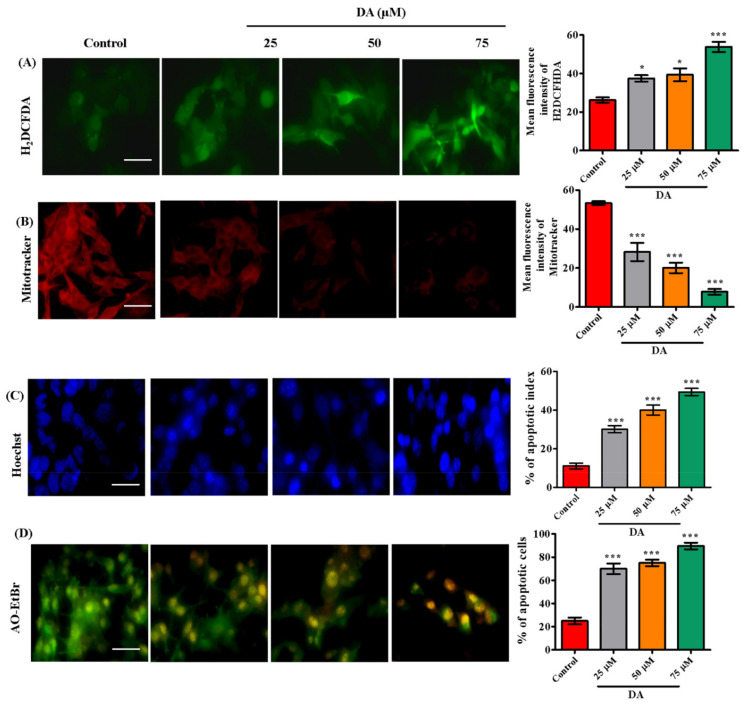
(**A**) The production of ROS was evaluated through H_2_DCHFDA (cellular ROS) on B16F10 cells after 24 h DA treatment. (**B**) Mitotracker staining to evaluate the mitochondrial membrane potential within live cells. (**C**) The apoptotic morphology and live-dead cells detected through Hoechst staining. (**D**) AO–EtBr staining. Quantification was done with the help of Image J software. Apoptotic cells were calculated as percent apoptotic nuclei compared with a total number of cells. Images were taken at 200× magnification using an Olympus BX50 fluorescence microscope (scale bar = 100 µm).

**Figure 6 ijms-22-04096-f006:**
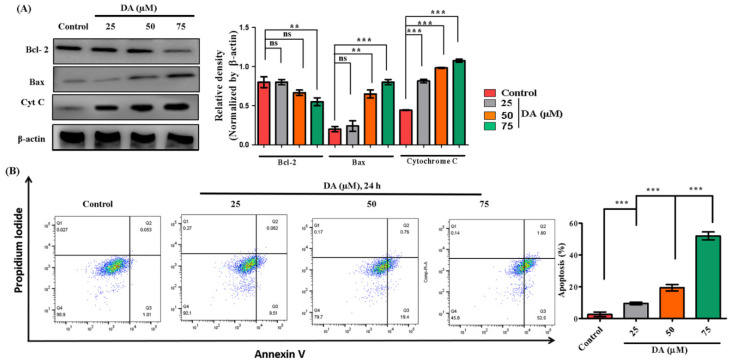
DA (25–75 μg/mL) induces apoptosis in murine melanoma B16F10 cells via the mitochondrial-mediated pathway. (**A**) Evaluation of apoptotic protein levels (Bcl-2, Bax and cytochrome C) by Western blotting. (**B**) Apoptotic occurrence in B16F10 cells analyzed by flow cytometry. Q1—necrosis, Q2—late apoptosis, Q3—early apoptosis, Q4—live healthy cells. The data are represented as the means ± S.D. of three independent experiments ** *p* < 0.01, *** *p* < 0.001 vs. control, calculated through ANOVA prism.

**Figure 7 ijms-22-04096-f007:**
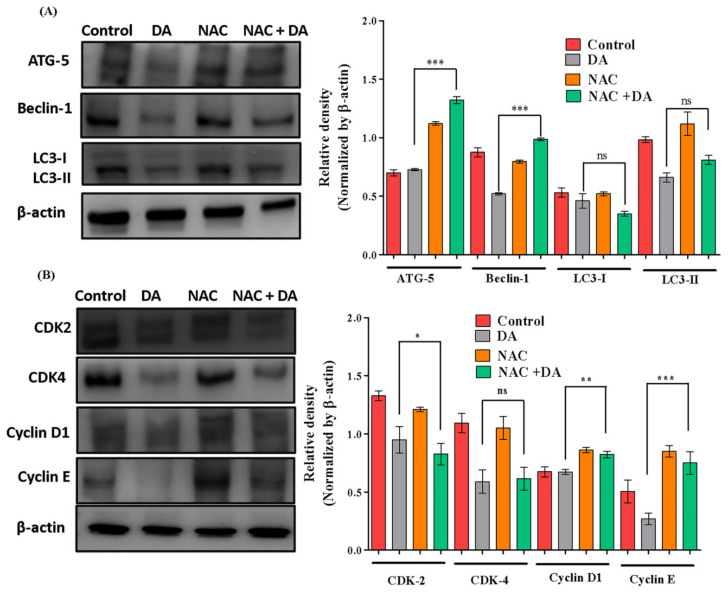
NAC treatment rescues autophagy inhibition by DA and G1 cell arrest after DA treatment. (**A**) Western blotting of important autophagy pathway markers, β-actin used as loading control. (**B**) Western blotting of important cell cycle pathway markers, β-actin used as loading control. Densitometry analysis was determined by Image J software. The data are represented as the means ± S.D. of three independent experiments * *p* < 0.05, ** *p* < 0.01, *** *p* < 0.001 DA vs. NAC + DA, calculated through ANOVA prism.

**Figure 8 ijms-22-04096-f008:**
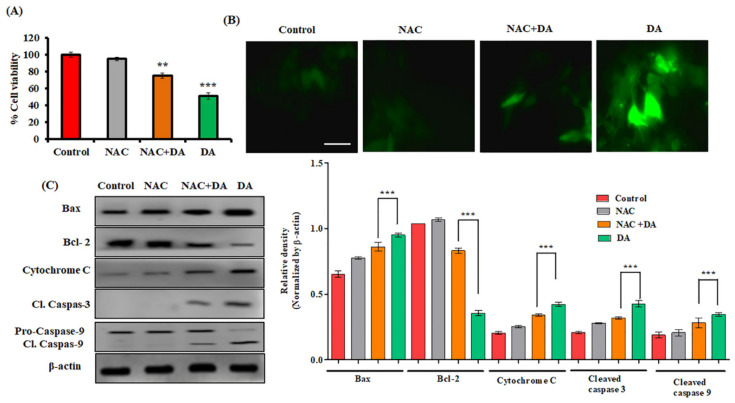
NAC mitigates the DA-induced mitochondrial-mediated apoptosis in B16F10 cells. (**A**) Effects of ROS scavenger NAC on DA-induced cell death evaluated by the MTT assay. Cells were exposed to NAC (5 mM, 3 h) and DA (75 μM, 24 h). (**B**) DA-induced ROS production abolished H_2_DCFDA fluorescence in the presence of NAC. Images were taken at 200× magnification (scale bar = 100 µm). (**C**) Western blots of apoptosis pathway markers; β-actin was used as loading control. Densitometry analysis was determined by Image J software. The data are represented as the means ± S.D. of three independent experiments ** *p* < 0.01, *** *p* < 0.001 NAC +DA vs. DA, calculated through ANOVA prism.

**Figure 9 ijms-22-04096-f009:**
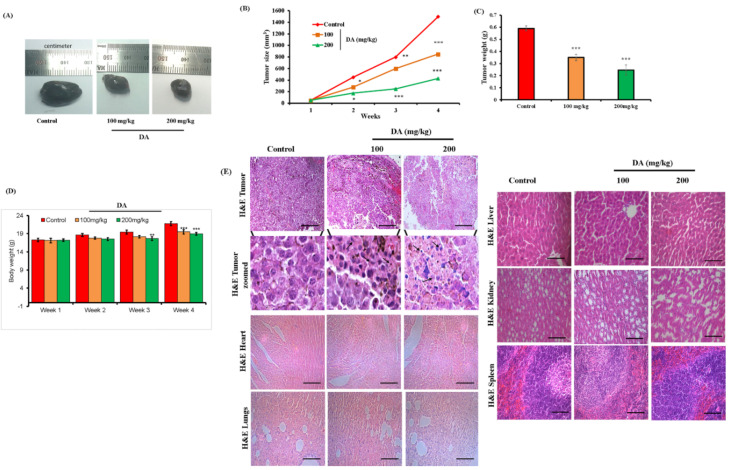
In vivo study: DA inhibits tumor growth. (**A**) Excised tumor images from different groups of mice after 4 weeks. (**B**) Tumor volume with respect to time. (**C**) Tumor weight after 4 weeks. (**D**) Body weight with respect to time. (**E**) H&E staining of tumor and other vital organs such as heart, lungs, liver, kidney and spleen (scale bar = 100 µm). arrow marks: represent the occurrence of apoptotic cells in the B16F10 tumors from H&E staining. The data are represented as the means ± S.D. of three independent experiments * *p* < 0.05, ** *p* < 0.01, *** *p* < 0.001 vs. control, calculated through ANOVA prism.

**Figure 10 ijms-22-04096-f010:**
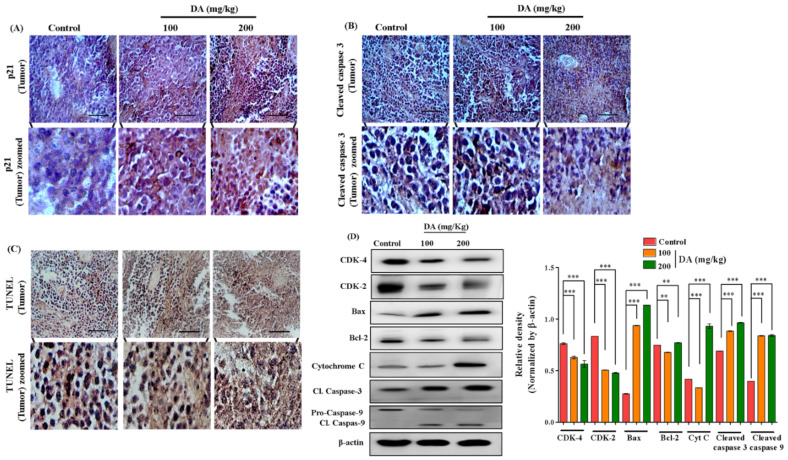
Histological images. (**A**) Immunohistochemical staining for p21 on B16F10 mice tumor. (**B**) Immunohistochemical staining was performed on B16F10 mice tumor for detection of cleaved caspase 3. (**C**) TUNEL assay on mice tumor. Images were captured at 200× magnification using an Olympus BX50 fluorescence microscope (scale bar = 100 µm). (**D**) Western blotting analysis for expression of CDK-4, CDK-2, caspase-9, caspase-3, cytochrome C, Bcl-2, Bax, β-actin in tumor tissue. Each value in the bar graph represents the mean ± S.D. of three independent experiments where ** *p* < 0.01, *** *p* < 0.001 vs. control, calculated through ANOVA prism.

**Figure 11 ijms-22-04096-f011:**
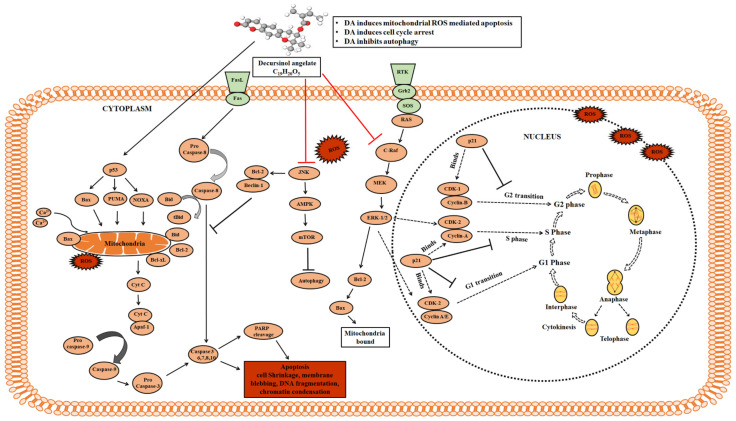
Schematic graphical representation of the proposed mechanism. Apoptotic effect of DA on B16F10 murine melanoma cancer cells. DA triggers the activation of mitochondrial apoptosis, subsequently resulting in apoptotic cell death.

## Data Availability

The authors declare that all the data supporting the finding of this study are available within the article and from the corresponding author on reasonable request.

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
