# Peer review of "Decursinol Angelate Arrest Melanoma Cell Proliferation by Initiating Cell Death and Tumor Shrinkage via Induction of Apoptosis"

_ijms, 2021, doi:10.3390/ijms22084096_

Round 1

Reviewer 1 Report

  1. According to the results presented in figure 8 (A, B, C), the densities of the tumor tissue differ very much, if we compare for example the control group with the group treated with 200 mg / kg DA. How can the authors explain these differences in the density of the studied tumor tissues?
  2. Also, from the presented data it results, for example, that the density of the tumor tissue of the control group is lower than the density of the water. This means that the tumors should float in the fixation medium. The authors can confirm whether this deduction I made in connection with the data they presented about the studied tumor tissue is true.
  3. In Figure 8E, microscopic features of several viscera are shown. What is the connection with this study of the microscopic aspects presented? The magnification of the images presented by the sections through tumors must be increased in order to be visible the morphological traits of the tumor tissue.
  4. The histological images from the control group, presented in figures 9 A, B, C, show different tumor areas: tumor without necrosis in 9A, respectively tumor necrosis areas with tumor cell nests in 9B and 9C. Considering the inherent necrosis in the 1.5 cm melanoma tumors, I think the same types of tumor regions should be presented. Also, can the authors explain how they took into account this tumor necrosis in the interpretation of the data obtained from the treated groups? The magnification of the images presented must be increased.

  1. The chapter on discussions must be reconsidered. Many paragraphs I think should be moved to the introductory chapter.
  2. NAC must be explained at the first appearance in the text (abstract)

Author Response

  • Author response to the comments of Reviewer 1

General Remark: Comments and Suggestions for Authors

Query 1: According to the results presented in figure 8 (A, B, C), the densities of the tumor tissue differ very much, if we compare for example the control group with the group treated with 200 mg / kg DA. How can the authors explain these differences in the density of the studied tumor tissues?

Author Response: Thank you for your insightful comments Sir. The densities of the tumor tissues vary significantly because of the treatment of decursinol angelate at (100 mg/kg and 200 mg/kg) in comparison to control (vehicle) which resulted in tumor size (Fig. 8A, revised to Fig. 9A), tumor volume (Fig. 8B, revised to Fig.9B) and tumor weight (Fig. 9C). we have also explained the phenomenon for the differences in the tumor being due to the altering cellular signaling pathway in the tumor tissues and we have depicted via H&E staining (Fig.9E) marking the regions of the occurrence of apoptosis. (Line number 208-210 in the manuscript).

Query 2: Also, from the presented data it results, for example, that the density of the tumor tissue of the control group is lower than the density of the water. This means that the tumors should float in the fixation medium. The authors can confirm whether this deduction I made in connection with the data they presented about the studied tumor tissue is true.

Author Response: Thank you for your insightful comments on the matter Sir. In the figure 9B, (Line number 203 in the manuscript). we have measured the weight of different tumor groups which were; control (0.59 grams), DA 100 mg/kg (0.35 grams) and DA 200 mg/kg (0.245 grams). The tumors sunk in the fixation medium i.e. 10% neutral buffer formalin solution for more than 1 day before we proceeded with the tissue processing for H&E and immunohistochemical staining. The tumors did not float in the fixation medium. All tumor groups were well fixed and settled under the fixation medium at same time (and were not floating in the fixation medium).

Query 3: In Figure 8E, microscopic features of several viscera are shown. What is the connection with this study of the microscopic aspects presented? The magnification of the images presented by the sections through tumors must be increased in order to be visible the morphological traits of the tumor tissue.

Author Response: Thank you for your insightful comments on the matter Sir. We have provided the explanation for the connection for the microscopic features of tumor with respect to the occurrence of apoptosis. Through H&E staining in figure. 8E (revised to figure 9E), we observed there was more occurrence of apoptotic cells in DA treatment groups in comparison to control (no DA treatment). We also observed the effect of DA on different organs of tumor bearing mice and found that the organs such as heart, lungs, liver, kidney and spleen showed no morphological damage or cellular distortion and were normal. (Line number 208-210). As per your suggestion, we have changed the H&E tumor images and also provided a cellular distortions and occurrences of apoptotic cells in zoomed tumor images labelled with arrow marks (Line number 202).

Query 4: The histological images from the control group, presented in figures 9 A, B, C, show different tumor areas: tumor without necrosis in 9A, respectively tumor necrosis areas with tumor cell nests in 9B and 9C. Considering the inherent necrosis in the 1.5 cm melanoma tumors, I think the same types of tumor regions should be presented. Also, can the authors explain how they took into account this tumor necrosis in the interpretation of the data obtained from the treated groups? The magnification of the images presented must be increased.

Author Response: Thank you for your insightful comments Sir. As per your suggestion, we have added a zoomed imaged for better observation of the immunohistochemical staining along with the original staining images for Figure 9 A, B and C (revised to Figure 10 A, B, C). Also, we have replaced the control image for Fig. 10A which show necrotic and apoptotic regions as well. Our focus of the study was on the occurrence of apoptosis and so we did not evaluate the markers for necrosis, however we are in agreement for the fact that the tumor shrinkage and cell death were caused by apoptotic occurrences and necrosis.  (Line number 220)

Query 5: The chapter on discussions must be reconsidered. Many paragraphs I think should be moved to the introductory chapter.

Author Response: Thank you for your insightful comments on the matter Sir. As per your suggestion we have shifted two paragraphs explaining the occurrence of cell cycle events and autophagy in the introductory section. (Line number 61-79)

Query 6 : NAC must be explained at the first appearance in the text (abstract)

Author Response: Thank you for your insightful comments on the matter sir. As per your suggestion we have added a more detailed explanation about the role of NAC in rescuing the cell cycle arrest, autophagosome inhibition and apoptosis induced by DA in the abstract. (Line number 26-32). However, since we carried out the experiment of DA treatment in B16F10 cells first and later NAC evaluation for its role in rescuing and ameliorating the cellular stress and cell death caused by DA, so sequentially we have mentioned about NAC treatment in the later sentences after DA treatment in the abstract.

Reviewer 2 Report

The authors presented the paper entitled "Decursinol angelate arrest B16F10 cell proliferation by initiating cell death and tumor shrinkage via induction of apoptosis" for the peer-review. Common chemotherapeutics used for treating cancers exert severe side effects and have variable efficiency. One of the promising strategies is the usage of different phytochemicals with a wide
range of pharmacological properties. Many studies have investigated the role of DA and its isomers anti-cancer activity. The present manuscript is well-written and structured. Methods are sound and relevant. The only suggestion I have is the option to use another cancer cell models to prove decursinol effects.

Author Response

  • Author response to the comments of Reviewer 2

General Remark: Comments and Suggestions for Authors

Query: The authors presented the paper entitled "Decursinol angelate arrest B16F10 cell proliferation by initiating cell death and tumor shrinkage via induction of apoptosis" for the peer-review. Common chemotherapeutics used for treating cancers exert severe side effects and have variable efficiency. One of the promising strategies is the usage of different phytochemicals with a wide range of pharmacological properties. Many studies have investigated the role of DA and its isomers anti-cancer activity. The present manuscript is well-written and structured. Methods are sound and relevant. The only suggestion I have is the option to use another cancer cell models to prove decursinol effects.

Author Response: Thank you for your insightful comments. As per your suggestion we have performed a preliminary study on human melanoma cell line (A375.SM cells)/. The datas are mentioned in Fig. 4A-D, (Line number 137-138, 152 in the manuscript). We have shown ROS staining, Hoechst for DNA damage and AO-EtBr for live and death cells. We also performed western blotting for early apoptosis markers for Bax, Bcl-2 and Cytochrome-C. Because of the limited time, we could not proceed further with major study involved in the manuscript, however, we wish to continue our study on the anticancer activity of DA in our next study. We sincerely hope that it will be acceptable to our respected reviewer.

Round 2

Reviewer 1 Report

I have no further comments to the authors

Author Response

Thank you for your time and consideration Sir. We greatly appreciate your insightful reviews into making the manuscript even better than initially submitted.